# Thermal and Electrical Properties of Additively Manufactured Polymer–Boron Nitride Composite

**DOI:** 10.3390/polym15051214

**Published:** 2023-02-28

**Authors:** Julia V. Bondareva, Daniil A. Chernodoubov, Oleg N. Dubinin, Andrey A. Tikhonov, Alexey P. Simonov, Nikolay V. Suetin, Mikhail A. Tarkhov, Zakhar I. Popov, Dmitry G. Kvashnin, Stanislav A. Evlashin, Alexander A. Safonov

**Affiliations:** 1Center for Materials Technologies, Skolkovo Institute of Science and Technology, 121205 Moscow, Russia; 2Emanuel Institute of Biochemical Physics of the Russian Academy of Sciences, 119334 Moscow, Russia; 3National Research Center “Kurchatov Institute”, 123182 Moscow, Russia; 4World-Class Research Center, Saint Petersburg State Marine Technical University, 190121 St. Petersburg, Russia; 5Skobeltsyn Institute of Nuclear Physics, Lomonosov Moscow State University, 119991 Moscow, Russia; 6Institute of Nanotechnology of Microelectronics of the Russian Academy of Sciences, 119991 Moscow, Russia; 7School of Chemistry and Technology of Polymer Materials, Plekhanov Russian University of Economics, Stremyanny Lane 36, 117997 Moscow, Russia

**Keywords:** photopolymer, boron nitride, thermal management, thermal and electrical conductivity, additive manufacturing, 3D-printed microelectronics

## Abstract

The efficiency of electronic microchip-based devices increases with advancements in technology, while their size decreases. This miniaturization leads to significant overheating of various electronic components, such as power transistors, processors, and power diodes, leading to a reduction in their lifespan and reliability. To address this issue, researchers are exploring the use of materials that offer efficient heat dissipation. One promising material is a polymer–boron nitride composite. This paper focuses on 3D printing using digital light processing of a model of a composite radiator with different boron nitride fillings. The measured absolute values of the thermal conductivity of such a composite in the temperature range of 3–300 K strongly depend on the concentration of boron nitride. Filling the photopolymer with boron nitride leads to a change in the behavior of the volt–current curves, which may be associated with the occurrence of percolation currents during the deposition of boron nitride. The ab initio calculations show the behavior and spatial orientation of BN flakes under the influence of an external electric field at the atomic level. These results demonstrate the potential use of photopolymer-based composite materials filled with boron nitride, which are manufactured using additive techniques, in modern electronics.

## 1. Introduction

One of the main problems in modern microelectronics is the overheating of devices and the resulting increase in the heat generated. If this generated heat is not effectively dissipated, the devices either fail or drastically reduce their efficiency. Due to the small size of microchips, it is difficult to use active cooling systems, such as coolants or fans; thus, finding suitable and inexpensive materials for thermal management is a real challenge [1]. One of the most common and cheapest materials for chip and circuit board packaging is polymers, which provide sealing, enhance mechanical protection, and prevent contact shorting. Since the mass fraction of polymer composites in microelectronic products is steadily increasing, developers are striving to adopt polymer materials for thermal management.

At first glance, polymeric structures do not withstand one of the main problems—insufficient heat dissipation caused by the reduced size of elements and, consequently, higher electric and thermal power density in and near the hot spots [2]. The low heat dissipation in polymers is due to the low thermal conductivity (k), which, for most polymers, lies in the range of 0.1–0.5 W m^−1^·K^−1^ [3,4,5,6]. Many studies have focused on ways to improve the thermal conductivity of polymeric materials, either by improving the alignment of polymer chains or by adding various fillers with a high thermal conductivity, such as carbon materials and various metal and ceramic particles [3]. The main problems arising from the filler addition method include finding the optimal filler material and loading ratio at which thermal conductivity increases but mechanical and electrical insulating properties are not degraded.

The ideal filling material should have the highest possible thermal conductivity and low electrical conductivity; in particular, pure semiconductor materials are suitable for this role because of the low concentration of free charge carriers and, therefore, low electrical conductivity and high thermal conductivity due to the contribution of phonons to the heat transfer. Additionally, the size and volume of the fractions must be taken into account when selecting a filler since, in micron-sized structures, the boundary scattering of low-frequency phonons leads to a decrease in the thermal conductivity value when compared to a large-sized bulk crystal, and the additive loading fraction can affect the thermal percolation threshold [7]. A high loading of fillers can cause problems with the uniform dispersion of the additives, as well as changes in viscosity and agglomeration, leading to increased complexity of fabrication and uneven distribution of properties, including thermal conductivity [8]. Moreover, in the case of photopolymer fillers, the absorption properties of the fillers can lead to partial absorption of radiation and alter the completeness of the polymerization reaction.

Based on the above reasons, boron nitride (BN) turns out to be one of the most optimal materials for thermal regulation [9,10,11]. Its hexagonal form has a graphite-like structure [12,13], where boron and nitrogen atoms are linked by strong covalent bonds within each layer, and the interlayer bonds are rather weak [14]. Such structure leads to a high k anisotropy—its value inside the basal plane reaches 420 W·m^−1^·K^−1^, and in the direction perpendicular to the plane, it is only about 5 W·m^−1^·K^−1^ [15]. The high k value inside the basal plane makes BN one of the best thermally conductive materials, especially among those where phonons dominate the thermal transport.

One of the widely used methods for fabricating complex three-dimensional structures from polymeric materials with various fillers is additive manufacturing (AM), which, due to its customization options, allows for design flexibility, cost-effectiveness, and relatively high production speed [16]. Several attempts have been made to use boron nitride nanosheets (BNNSs) embedded in thermoplastic polyurethane and polystyrene that are manufactured using fused deposition modeling (FDM) technology, and the thermal performance of these products has been investigated [17,18]. Another completely different method that is often used for 3D printing ceramic [19,20] and polymeric structures [21,22] is digital light processing (DLP), which has a faster printing speed than other methods and allows the production of designed structures with a relatively high accuracy of up to 20 μm [23,24,25]. The use of DLP technology, which is based on photopolymerization, requires careful consideration of viscosity when incorporating additives, such as boron nitride and others. The ideal viscosity of the starting printing material plays a crucial role in determining the maximum amount of the filler that can be added and, therefore, impacts the characteristics and properties of the printed samples.

This paper investigates the possibility of using DLP technology to print thermally conductive polymer materials for electronics. The samples of photopolymer with incorporated BN flakes were prepared using the DLP method. Coplanar interdigital capacitors coated with the same mixture of polymer and boron nitride were also fabricated to evaluate their current–voltage characteristics. Ab initio calculations were performed to show the behavior and spatial orientation of the BN flakes under the influence of an external electric field at the atomic level. We show for the first time that the addition of BN flakes has a significant effect on the thermal conductivity of the resulting composites, increasing it by a factor of two in the case of 20% of BN addition.

## 2. Materials and Methods

### 2.1. Materials

The photopolymer “Dental Yellow Clear” (HARZ Labs, Moscow, Russia) consists of (meth)acrylated monomers and oligomers with a photoinitiator and hexagonal BN flakes (RusChem, St Petersburg, Russia) with a particle size distribution from 5 to 20 μm.

### 2.2. Methods

#### 2.2.1. Photopolymer Preparation

Two types of samples were printed to compare thermal conductivity: pure photopolymer sample and photopolymer samples with different BN volume fraction (15 vol.% and 20 vol.%). When the concentration exceeds 20 vol.%, there is a significant increase in solution viscosity, which makes printing impossible. The photopolymer we used was commercially available “Dental Yellow Clear” produced by HARZ Labs (Russia). This photopolymer material is designed for use in SLA and DLP 3D printers with emission wavelengths from 350 to 465 nm. The photopolymer Dental Yellow Clear (HARZ Labs, Russia) was mixed with hexagonal BN flakes (RusChem, Russia). Mixing was performed on a SpeedMixer DAC 150 planetary mixer (Hauschild, Hamm, Germany) at 2000 rpm for 2 min.

#### 2.2.2. 3D Printing

Printing was performed on an AnyCubic Photon (Anycubic Technology Co., Hongkong, China) DLP (Digital Light Processing) 3D printer equipped with an LCD display with 405 nm LED backlight. The thickness of one printed layer was set at 25 µm. The exposure time and vertical speed of the platform were set differently for printing the samples of pure photopolymer material and the samples with the filler. This is because the materials differ greatly in viscosity and polymerization depth. For the pure photopolymer material, the exposure time per layer was 4 s and the vertical platform travel speed was set to 80 mm/min. For the photopolymer material with BN, the layer exposure time was 15 s and the platform travel speed was 5 mm/min. The printed samples had a square cross section and dimensions of 2 × 2 × 20 mm^3^. Two samples were prepared with a BN volume fraction of 15% (BN 15%) and 20% (BN 20%). The bulk density of the materials was estimated using the Archimedes method on the Ohaus Pioneer PX84/E (USA) with a density measurement kit.

#### 2.2.3. Particle Morphology Analysis

The morphology and particles size of BN were studied using a scanning electron microscopy (SEM) Quattro S (Thermo Fisher Scientific Inc., Waltham, MA, USA). To obtain the statistics on particle sizes, the SEM images of the particles were analyzed using the ImageJ software, which allows the aggregation of the statistics on particle sizes from several images.

#### 2.2.4. Spectroscopic Characterization

The printed pure polymer and BN-filled samples were examined to trace the specific peaks attributed to different components using a Thermo Scientific DXR xi Raman imaging microscope with Thermo Scientific OMNIC xi Raman imaging software. The measurements were conducted at room temperature and at the excitation wavelength of 532 nm from a laser that focused on the samples using a 50× objective for the 100–3500 cm^−1^ zone. The obtained spectra were exported to the Origin Pro data analysis software for data processing. The baseline was corrected, and the peak heights were normalized to 1.0.

#### 2.2.5. Thermal Analysis

Thermal analysis of the printed pure polymer and BN-filled samples was carried out using a DTG-60 thermal analyzer (Shimadzu, Kyoto, Japan). The samples with a weight of 10 mg were placed into Al_2_O_3_ pans and exposed to heating at temperatures up to 900 °C, with a heating rate of 5 °C min^−1^, to evaluate the decomposition temperature (T_d_) under argon using a flow rate of about 80 mL min^−1^.

#### 2.2.6. Mechanical Test

In the three-point bending test of the polymer samples, the flexural strength of the polymer samples was determined using the universal testing machine Instron 5969 in accordance with the ASTM C 1167. The bending tests were performed on the samples that had a length of 45 mm, a width of 4 mm, and a thickness of 3 mm at room temperature. The roller diameter gauge length and the gauge length were 4 mm and 40 mm, respectively.

#### 2.2.7. Thermal Conductivity Measurements

The thermal conductivity measurement was conducted using the Quantum Design PPMS measurement system. The conductive copper wires were connected to the samples with the epoxy EPO-TEK H20E, such that the resulting distance between the thermometers, which were connected to the clamps, was 3 mm. The measurements were performed in the 3–300 K temperature range. The heating mapping was performed using a Fluke Ti450 (Fluke Corporation, Everett, WA, USA).

#### 2.2.8. Electrical Properties

The coplanar interdigital capacitors were made of thermal silicon oxide with an area of 7 × 7 mm^2^, a width of the individual pins of 100 microns, and a gap between the pins of 50 microns. The capacitor contacts were made of aluminum with a thickness of 150 nm. The polymers with and without boron nitride flakes of different concentrations were applied by using the calibrated drop method.

Capacitance was measured with a Keysight E4980 precision RLC meter using a Kelvin probe. The bias voltage was 10 mV, which is much less than the breakdown voltage. The measurements were made according to the two models, Cp-Rp and Cs-Rs, in the frequency range up to 300 kHz.

#### 2.2.9. DFT Calculations

Quantum chemical calculations were performed in the framework of density functional theory (DFT) using LCAO (DZP basis set) approximation implemented in the SIESTA software [26]. To describe the atomic structure of the BN flakes under investigation, modified GGA-PBE functional (revPBE) [27] was used. The energy mesh was set as 225 Ry, which provides a better convergence when calculating with an external electric field. Only gamma-point calculations were performed for the BN flakes in relaxation. The structural relaxation was carried out until the change in the total energy was less than 10^–5^ eV. To avoid artificial interactions between the structures in non-periodic directions, the distance of 20 Å was chosen. The lateral size of the flakes was 13.8 Å. The total number of atoms in the tested unit cells were 72 atoms (27 boron and nitrogen atoms and 18 hydrogens). We chose the simplest configuration with the aim of qualitatively describing the features of the external electric field (EEF) effect on the BN flakes’ spatial orientation. The external electric field was applied initially along the armchair direction of the BN flakes and had a magnitude of 0.5 V/Å to reveal the EEF’s effective influence.

## 3. Results and Discussion

### 3.1. Sample Preparation

Boron nitride in the volume fractions of 15 and 20% was added to a mixture of methylacrylate oligomers with a photoinitiator. The mixture was then stirred to achieve homogeneity. The percentage of boron nitride was carefully chosen to achieve the optimum viscosity of the mixture and a uniform distribution of boron nitride in the printing mixture. The resulting mixture was then used for 3D printing according to the method described in Section 2.2.2. Typical defects that can occur during DLP 3D printing include delamination and formation of air bubbles within the printed layers. To address the former issue, the amount of light exposure per layer is increased, and the printed layers are overlapped (in this study, there was double overlapping). To prevent the latter issue, defoamers are added to the photoresin composition used in the printing process. All printed samples were subjected to density measurements using Archimedes’ principle. The samples were immersed in an experimental test tube in which the water level was known in advance to determine its volume. The density values were then obtained after the mass of each sample was measured using analytical scales. The bulk density of the clean polymer and the polymers with 15 and 20% of BN were 1.21, 1.38, and 1.43 g/cm^3^, respectively. The printed materials were found to have some degree of porosity, which could potentially impact their mechanical and thermal properties [28]. Compared to other printing techniques, the DLP process typically results in lower material porosity due to its use of liquid polymers and the scattering of light, as explained in a previous study [29]. The calculated porosity for all types of samples was lower than 0.5%.

### 3.2. Particle Morphology Analysis

The morphology of the pristine BN powder was studied using SEM. As shown in Figure 1a, the pristine BN powder shows the presence of particles with a size of around 5–20 μm (Figure 1b), and plenty of BN sheets are stacked together.

### 3.3. Spectroscopic Characterization

The Raman spectrum of BN exhibits a strong absorption at 1365 cm^−1^, corresponding to B−N stretching (in-plane ring vibration), which persists in the BN-filled polymer sample (Figure 2). The Raman spectrum of the polymer shows that the band at 2935 cm^−1^, which is characteristic of PMMA-like polymers, indicates C–H stretching. The polymer sample also shows a peak at 1718 cm^−1^, which corresponds to ester carbonyl C=O stretching vibration. Another peak appears at 1450 cm^−1^, which corresponds to the C–H bending vibrations of the polymer skeleton. The band at 1640 cm^−1^ can be attributed to a combined band arising from n (C=C) and n (C–COO). There are other Raman bands, such as the band at 600 cm^−1^ is attributed to (C–COO), (C–C–O); the band at 833 cm^−1^ is due to CH_2_; the band at 1264 cm^−1^ is due to (C–O), (C–COO); the band at 1118 cm^−1^ is due to the (C–C) skeletal mode; and the band at 2879 cm^−1^ is due to the combination band involving O–CH_3_ [30,31].

### 3.4. Thermal Analysis

To evaluate the thermal stability of the composite samples with and without boron nitride, TGA was used from 20 °C to 600 °C, at a constant heating rate. Figure 3 shows the TGA curves for the composite samples. No significant difference is observed between the TGA curves for all samples and the clean polymer. The results reveal that the polymers loaded with BN are thermally more stable. Compared to the clean polymer, the filled samples start to degrade at higher temperatures, and the rates of weight loss decrease remarkably. Thus, the clean polymer starts to degrade at ~160 °C, while the polymer samples filled with BN degrade at ~214 °C. The enhancement of thermal stability may be attributed to the incorporation of BN flakes, which can inhibit the mobility of polymer chains and help capture free radicals generated during pyrolysis. Furthermore, all polymer and polymer-composite samples were found to degrade mostly through a two-step reaction. The thermal degradation occurs in two steps at the temperature ranges of 160–360 °C and 360–440 °C [32], for a heating rate of 5 °C/min. The first step is due to the degradation of thermally weaker bonds, terminal groups, and initiator residue situated in the polymer chain, while the second step can be attributed to random scission and chain depolymerization [33,34].

### 3.5. Mechanical Analysis

Figure 4 shows the results of the bending test conducted on the polymer samples. The sample without fillers exhibited an ultimate strength of 42 MPa and a ductility of approximately 12%. When boron nitride was added to the polymer, the ultimate strength of the samples increased, with a maximum value of 47 MPa. However, the ductility of the samples decreased to 2.5%. The yield strength of the samples varied, with values of 30 MPa, 35 MPa, and 40 MPa for the unfilled polymer and the polymers containing 15% and 20% BN, respectively. The Young’s modulus of the samples also changed from 1000 MPa to 3000 MPa.

### 3.6. Thermal Conductivity Measurements

Thermal conductivity in a composite system is governed by phonons, i.e., quasiparticles corresponding to the lattice vibrations. Researchers can use the following formula to understand phonons’ thermal conductivity behavior [35]:$$k=\frac{1}{3}C_{\nu }\nu_{c}\Lambda$$
where *Cv* is the lattice specific heat per unit volume, *v* is the sound velocity, and Λ is a phonon mean free path. In general, in pure semiconductor crystals and in the limit of low temperatures, the phonon mean free path is limited by the crystal boundaries; therefore, its thermal conductivity varies as $\kappa \propto C_{v}\propto T^{3}$. At higher temperatures, where more phonons of higher frequencies (therefore, having lower wavelengths) start to take part in thermal conductance, the thermal conductivity behavior becomes more complex because the phonon mean free paths start to be defined by processes other than the boundary scattering processes, such as scattering on defects and phonon–phonon scattering.

In amorphous media, such as glasses or polymers, the disorder at the molecular level is so high that mean free path is effectively constantly limited in a big temperature range and is roughly proportional to the heat capacity alone, being $\kappa \propto C_{v}\propto T^{3}$ at low temperatures and $\kappa \propto C_{v}\propto T$ at high temperatures. At the same time, there exists a temperature region at intermediate temperatures where thermal transport starts to be dominated by phonons with wavelengths large enough to not be scattered on structural heterogeneities. This leads to a thermal conductivity plateau [36].

Thermal conductivity of a composite can be explained on the basis of the effective medium theory [37]. The resulting thermal conductivity of a composite depends on the thermal conductivities of its components, their fraction, the thermal boundary resistances between them, and the particle shapes. There are more than a dozen of composite thermal conductivity models that take into account these factors [38]. Thermal boundary resistance is an important factor for understanding thermal transport in composites due to the possible high acoustic mismatch between the components of a composite [39].

Thermal conductivity was measured for all samples in the range of 3–300 K (Figure 5). The addition of BN flakes has a significant effect on the thermal conductivity of the resulting composites, increasing it by a factor of two at room temperature in the case of the 20% BN-filled sample. The obtained results are in accordance with previous measurements for polymer/semiconductor composites, e.g., [40].

The introduction of BN flakes into a polymer results in two-sided action: on the one hand, the addition of a higher conductive phase should lead to an increase in thermal conductivity in accordance with the effective medium theory; however, on the other hand, the flakes become an additional obstacle for the heat carriers in the host medium, introducing additional thermal boundary resistance and limiting the mean free path [41], which leads to lower values of thermal conductivity at lower temperatures. Both of these effects are seen in the obtained data—at high temperatures, the samples with added flakes have a significantly higher thermal conductivity value, but in the limit of low temperatures, their thermal conductivity value is lower than that of the pure sample due to the mean free paths being limited by the flakes and the thermal boundary resistance between the BN particles and the polymer. The observed behavior, showing that when at temperatures below 20 K, the thermal conductivity of a polymer composite is lower than that of a pure polymer, has been observed previously for added particles with the sizes of about 10 µm [40].

To evaluate the thermophysical properties of composites with and without filler, the models of radiators made of pure polymer and a composite with 20% boron nitride filling were printed. Both printed models (from pure photopolymer material and from a mixture of photopolymer and boron nitride powder) were placed on the heating surface of a laboratory hotplate PLP-03 (Tomanalit, Russia) (Figure 6a). The hot plate was turned on to 100 °C with the maximum speed. On the bottom surface of the models, a thin layer of thermal conductive paste KPT-8 was placed in advance for more uniform contact of the bottom side plane of the samples with the plane of the heating hot plate.

One minute after the hotplate was heated to 100 °C, an image was captured with a Fluke Ti450 thermal imager (Fluke Corporation, USA). The image shows two points with the temperature at the same height from the base of the samples, corresponding to a sample of the BN-filled photopolymer and a sample of the pure photopolymer material (Figure 6b). The temperature at the point set in the middle of the side of the BN-filled sample is 57.9 °C, and the temperature at a similar point of the clean polymer sample is 40.7 °C. The thermal image also shows that the heating of the sample with boron nitride is more uniform.

### 3.7. Electrical Properties

Electrical properties were measured with AC and DC. Figure 7a shows the composite samples’ measurement scheme. A photograph of the structures is shown in Figure 7b. During the AC measurements, the frequency varied from 20 Hz to 300 kHz. Two AC measurement schemes were employed—the first equivalent circuit is a parallel connection of a resistor and a capacitor C_p_ and R_p_, and the second is a series connection of C_s_ and R_s_, as shown in Figure 7. For both cases, an increase in the capacity of the material filled with BN is observed. At the same time, at high frequencies, an almost linear dependence is observed. The obtained resistance value decreases in the case of added BN flakes, as shown in Figure 7d. Such a decrease in resistance is most likely due to the percolation currents that arise due to the deposition of the BN material. At the same time, in the series connection scheme, the values of the filled and pure polymers differ less.

A plot of resistance versus voltage obtained with the DC measurements is shown in Figure 8. All polymers show an increase in resistance with small voltage application, but then there is a significant decrease. Such a change is apparently associated with the presence of monomers and impurities in the material, which can lead to polarization. As is known, not all monomers are polymerized in photopolymers. At the same time, it should be noted that the field value reaches tens of volts per micron.

Filling the photopolymer with boron nitride leads to a change in the behavior of the curve. Similar to the pure polymer, an increase in resistance is observed, but it is less significant. Then, as the voltage increases, the resistance increases. This increase in resistance can be explained by the presence of boron nitride, which aims to line up in the field, as shown in the DFT calculations. This alignment leads to an increase in resistance. 

### 3.8. DFT Calculations

The DFT can be a powerful tool to reveal an EEF’s influence on a structure at the atomic level. In Figure 9, we present the results of the investigation of EEF influence on BN flakes and their spatial orientation. The atomic forces acting on each atom of BN flakes under the EEF depends on their spatial orientation in the XY plane, as presented in Figure 9a. We rotated the BN flakes with a step of 15°. A dominant orientation of forces along the EEF direction was observed. It was found that applying an EEF to the BN flakes led to a change in the total dipole moment of the flakes with respect to the changes in the total energy. In Figure 9b, the dependence of the y-component of the dipole moment (D_y_, red dots) and total energy difference (blue dots) on the rotation angle is presented. It was found that the minimum total energy of the BN flakes under the applied EEF corresponds to 60° of rotation angle (armchair direction of BN flakes) with a maximum y-component of the dipole moment. It is important that the difference between the atomic structures of the BN flakes with 0° and 60° of rotation angle is the edge termination along the EEF direction. In the case of N-B termination (60°), the D_y_ is maximum, while the B-N termination leads to a minimal value of D_y_. In Figure 9c, the dependence of the xy components of the mechanical stress tensor is presented. Two minimums at 0° and 60° are found, which show the stationary spatial orientation of the BN flakes within the applied EEF. The obtained results show that under the action of the EEF, BN flakes tend to orient along the EEF direction.

The same calculations were performed to study the influence of the EEF during the rotation of the BN flakes in the yz plane (Figure 10). Here, we rotated the flakes with a step of 30°. The atomic force calculations show a dominant orientation along the applied EEF direction. The calculation of the total energy of the BN flakes depends on the rotation angle, showing two minimums at 0 and 180, which correspond with the maximums of the dipole moment components (D_y_ and D_z_ in red circles and rectangles, respectively). Based on the obtained results, we can state that under the EEF, BN flakes tend to rotate and align in the armchair direction along the EEF. This can qualitatively explain the experimentally observed nonlinear behavior of the XZ curves.

## 4. Conclusions

The objective of this study was to examine the feasibility of utilizing the DLP technique to fabricate thermally conductive polymer composite material. The addition of boron nitride aimed to enhance the thermal conductivity of the polymer material, which is crucial for optimal heat transfer and dissipation in microelectronic devices. Three types of printed samples (0 vol.%, 15 vol.%, and 20 vol.%) were produced and analyzed using thermogravimetry and Raman spectroscopy. A maximum volume fraction of 20% BN was limited by the viscosity of the solution. The higher concentration limited the printability of the DLP technology. The addition of BN in the photopolymer influenced the mechanical, thermal, and electrical properties. The ultimate strength of the pure polymer and the BN-filled polymer was 42 and 47 MPa, respectively. The ductility of the materials dropped from 12 to 2.5 %. The thermal conductivity measurements revealed a strong correlation between the thermal conductivity of the samples and the volume fraction of BN. The sample filled with 20% boron nitride showed a higher thermal conductivity and uniform heat distribution. The thermal conductivity of the materials at room temperature increased by a factor of two. Additionally, the volt–current characteristics of the printed samples were evaluated, revealing an increase in capacitance and a decrease in resistance for the samples filled with boron nitride, which might be attributed to percolation currents caused by the deposition of boron nitride. The ab initio calculations aligned with our experimental observations and showed that the physical properties of BN flakes were dependent on the orientation of the external electric field. The outcomes of this study suggest that the DLP method is a suitable option for 3D printing of composite materials with the inclusion of boron nitride or similar fillers to improve the thermal properties of polymeric materials, thereby providing opportunities for the development of modern electronic devices with efficient heat management.

## Figures and Tables

**Figure 1 polymers-15-01214-f001:**
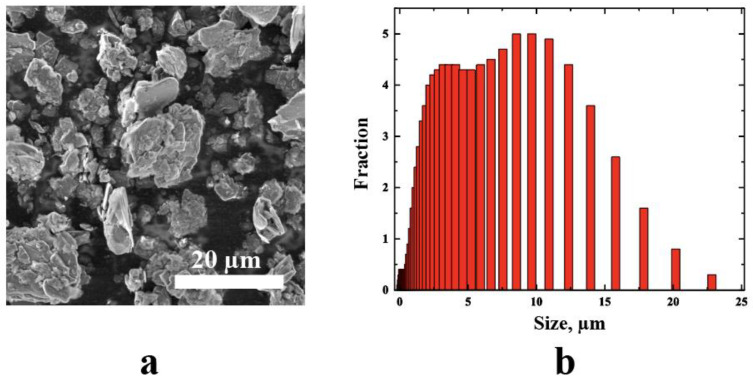
(**a**) SEM image of hBN particles, and (**b**) particle size distribution (PSD) histogram.

**Figure 2 polymers-15-01214-f002:**
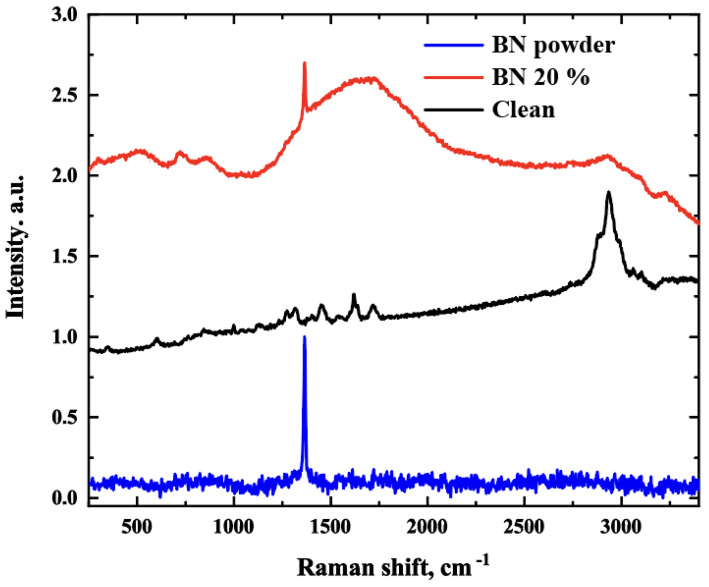
Raman spectra of BN powder, clean polymer, and 20% BN–filled polymer.

**Figure 3 polymers-15-01214-f003:**
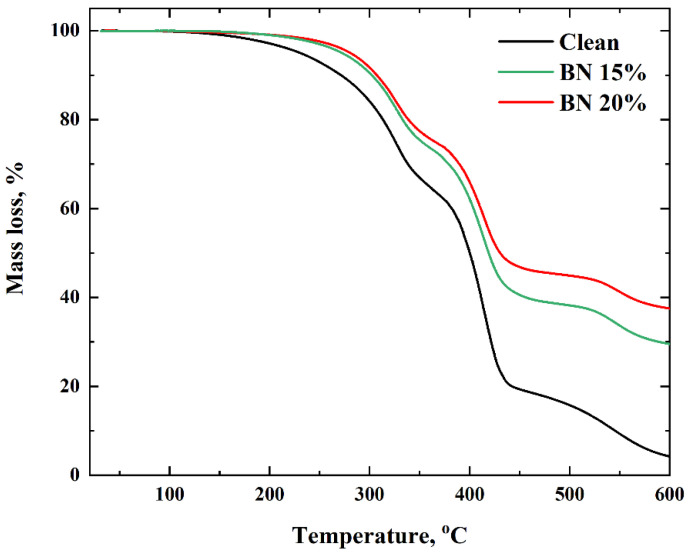
TGA analysis of the samples.

**Figure 4 polymers-15-01214-f004:**
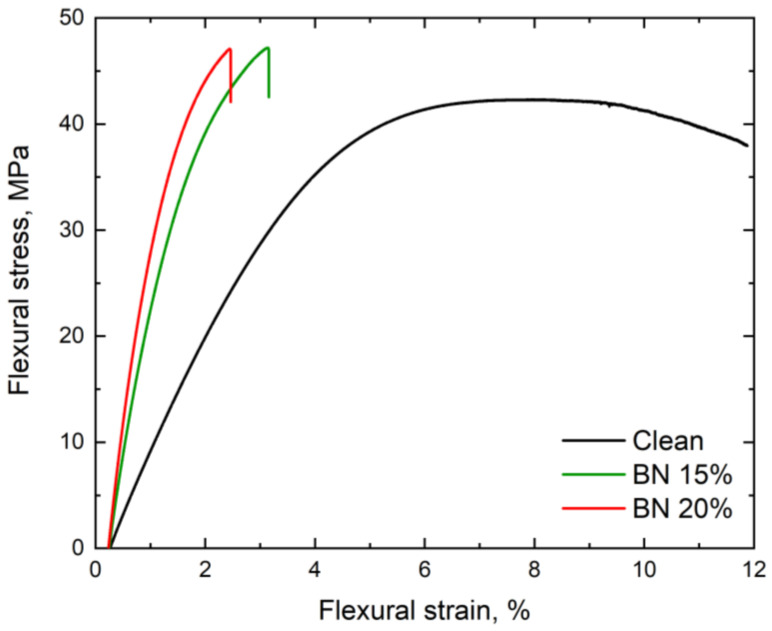
Flexural stress–strain curve for the clean polymer and the polymer composites with 15% and 20% BN.

**Figure 5 polymers-15-01214-f005:**
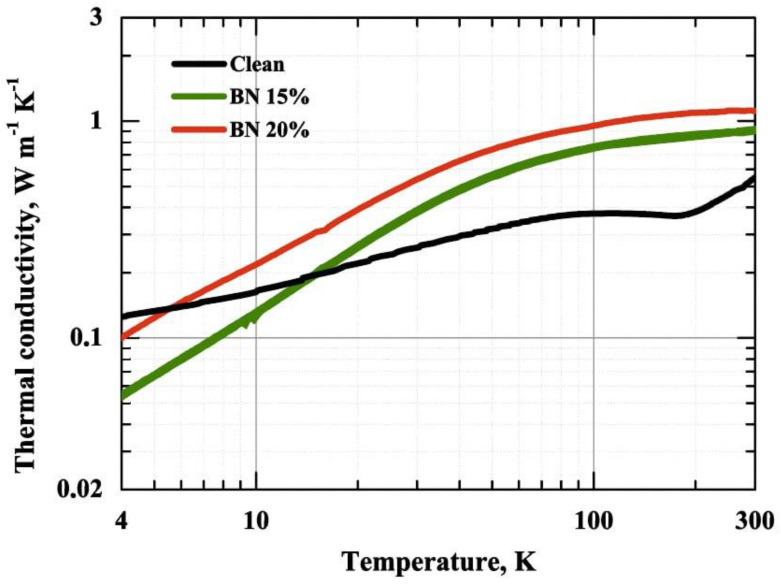
Thermal conductivity measurements for the pure polymer and the BN–filled polymers.

**Figure 6 polymers-15-01214-f006:**
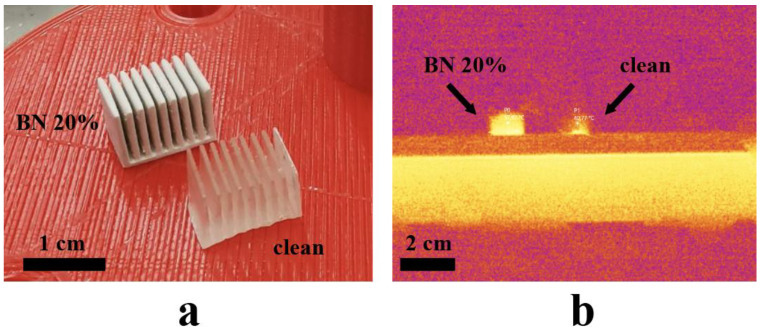
(**a**) Printed samples with and without BN, and (**b**) heat map of the heating process of the two samples.

**Figure 7 polymers-15-01214-f007:**
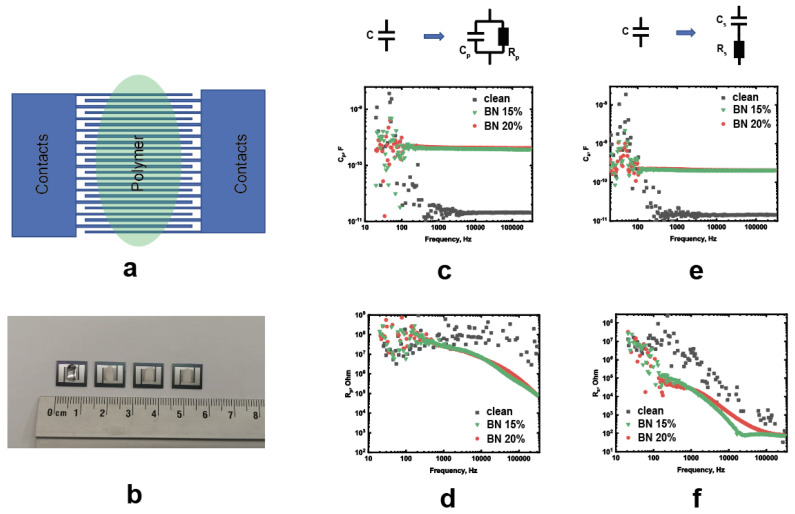
(**a**) Scheme of the coplanar interdigital capacitors with the composite sample; (**b**) photograph of the samples; (**c**,**d**) frequency-dependent C_p_ and R_p_ for the parallel connection; and (**e**,**f**) frequency-dependent C_p_ and Rp for the series connection.

**Figure 8 polymers-15-01214-f008:**
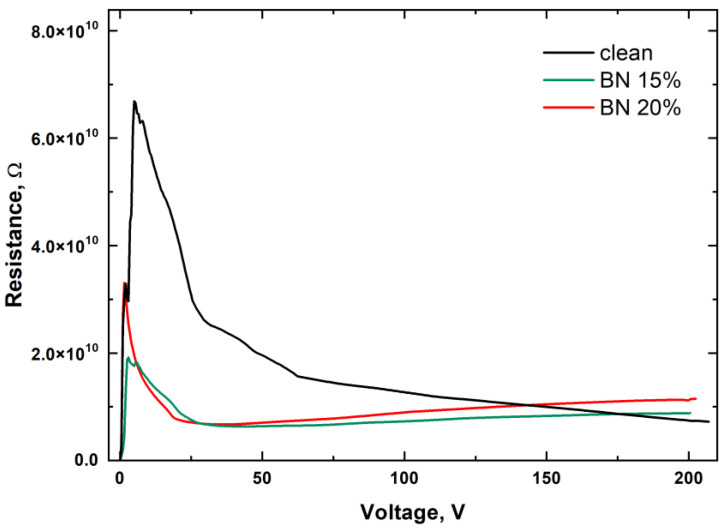
Resistance versus voltage obtained with the DC measurements.

**Figure 9 polymers-15-01214-f009:**
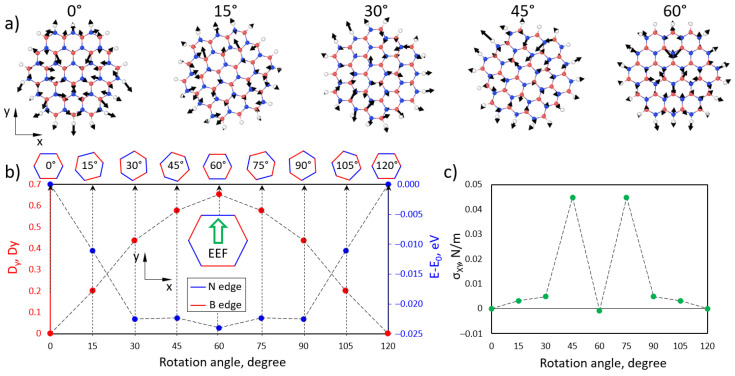
(**a**) Atomic structure of BN flakes with calculated atomic forces (black arrows) under the influence of the EEF. (**b**) Dependence of the y-component of the BN flake dipole moment (red dots) and total energy (blue dots) on the rotation angle in the xy direction. (**c**) Dependence of the xy components of the mechanical stress tensor on the rotation angle.

**Figure 10 polymers-15-01214-f010:**
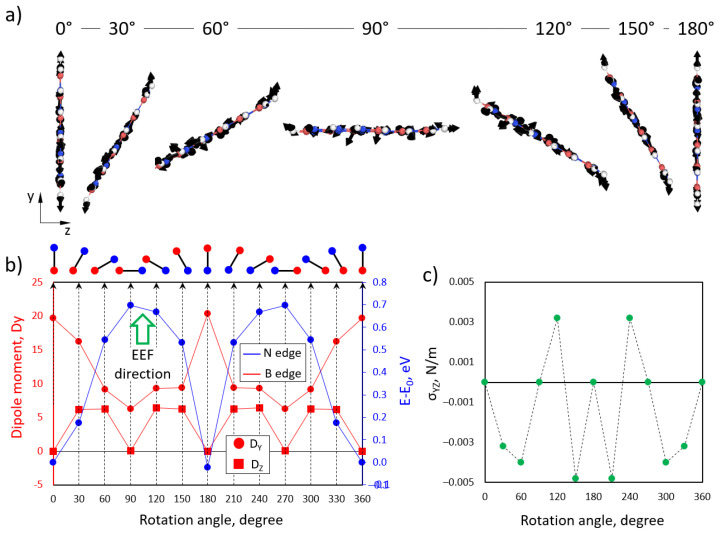
(**a**) Side view of the atomic structure of BN flakes with calculated atomic forces (black arrows) under the influence of an EEF. (**b**) Dependence of the y-component (red circles) and the component (red rectangles) of the BN flake dipole moment and total energy (blue dots) on the rotation angle in the yz direction. (**c**) Dependence of the yz components of the mechanical stress tensor on the rotation angle.

## Data Availability

Not applicable.

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
