# Peer review of "Thermal and Electrical Properties of Additively Manufactured Polymer–Boron Nitride Composite"

_polymers, 2023, doi:10.3390/polym15051214_

Round 1
Reviewer 1 Report
The following is my comments:
(1) The strategy of using Boron nitride for the enhancement of thermal management in polymer compound is well-known. There are tens of publications related with this subject.
The authors are requested to discuss the novelty of their work.
(2) The reason of using BN 20% rather than other concentration may have to be pointed out.
(3) The exact molecular structure of the polymer used has to be mentioned.
(4)The specific band in the Raman spectrum can be related with specific group structure of the polymer. This may have to be discussed.
Reviewer 2 Report
Overheating in microelectronics is a significant problem that still owes much research. I appreciate the author's effort in the additive manufacturing photopolymer-boron nitride composites. I would like to accept the paper with minor revisions.
Minor comments:
- In section 2.1, Materials, The author mentioned the particle size distribution from 0 - 20 um, which needs to be corrected since the author said the particle size to 5-20 um in the structural characterization(title of this section can be changed to particle morphology analysis) section.
- In section 3.2, Spectroscopic characterization, Further explanation is necessary. The author mentioned all the peaks are consistent with the literature without citing any work. It would be helpful if the author could explain what each peak corresponds to and what information is gained from it with proper reference.
- In section 3.3, Thermal analysis, The author studied the material's thermal stability using TGA. The author concluded that no significant difference was observed in TGA, which appears to be an incorrect statement. There is a significant difference observed in the curves. The onset of decomposition temperature increases with an increase in the Boron nitride volume fraction. The author should mention and comment on the decomposition temperature. There are multiple reaction steps during decomposition; address all of them.
- In section 3.4, Thermal conductivity measurements, The author investigated the influence of boron nitride volume fraction for temperature 3 - 300 K. Thermal conductivity in polymers is attributed to phonon transport. The author needs to make further explanations and a literature review regarding the phonon transport in polymers and interface boundary scattering of phonons which might also contribute to the reduction in thermal conductivity. The mean free path of phonon is extremely low at low temperatures. The specific heat capacity of boron nitride and polymers might give more insights into the mechanism behind the lower thermal conductivity of BN composite at low temperatures.
Reviewer 3 Report
The subject is timely and of great interest. The authors presented digital light processing for composite radiator with different boron nitride filling. They found that the absolute values of thermal conductivity strongly depend on the concentration of boron nitride. The results are interesting and the discussion is of in depth. The reviewer believes this is a paper of high quality and it is suggested that the article be accepted after the following minor comments are taken care of. I will be happy to review the revised manuscript again.
1. Additive/3D printing polymers (and other materials) often generates defects. Are there defects occurring during printing? Can you in situ detect the defects such as pores in the parts? The authors need to talk about such aspect with reference to the following paper.
In situ real time defect detection of 3D printed parts, Additive Manufacturing, 17 (2017) 135-142
2. BN particles have exceptional strength and other properties. What mechanical strength does this additively manufactured composite strong like other boron based composites. The authors need to talk about such aspect and compare with other boron based composites with reference to the following papers.
In situ Nanomechanical Characterization of Single‐Crystalline Boron Nanowires by Buckling, Small, 6 (2010) 927-931
B4C‐Nanowires/Carbon‐Microfiber Hybrid Structures and Composites from Cotton T‐shirts, Advanced Materials, 22 (2010) 2055-2059
B/SiOx Nanonecklace Reinforced Nanocomposites by Unique Mechanical Interlocking Mechanism, Advanced Materials, 20 (2008) 4091-4096
3. Other nano reinforcements Like TiC and SiC ad B4C also show great potentials to reinforce polymers. Can the methodology used in this paper be applied to other nano reinforcements? The authors need to extend the discussion with reference to the papers below.
TaC nanowire/activated carbon microfiber hybrid structures from bamboo fibers, Advanced Energy Materials, 1 (2011) 534-539
A generic bamboo-based carbothermal method for preparing carbide (SiC, B4C, TiC, TaC, NbC, TixNb1− xC, and TaxNb1− xC) nanowires, Journal of Materials Chemistry, 21 (25), 9095-9102
Reviewer 4 Report
Reviewer Comment for Editor/Editor-in-Chief:
The present manuscript presents a study regarding the possibility of using digital light processing (DLP) technology to print thermally conductive polymer materials for electronics. The samples of photopolymer with incorporated BN flakes were prepared by the DLP method.
This manuscript could potentially be suitable for publication, but it needs a minor revision before it could be published.
1. There is a wrong affiliation number (i.e. 2 numbers are 4). Please correct.
2. In the abstract there are mixing of present and past form of tenses. Such kind of mixing should be avoided. It should be one particular form.
3. The thermal conductivity symbol (k) should be unified in the whole manuscript i.e. page 2, line 23.
4. In the section (2.2.2. 3D printing), please correct 2x2x20 by using “×” instead. Same in part 2.2.7.
5. The results and discussion section should be started with one paragraph describing the polymer sample synthesis method.
6. In figure 2, the sample names must be same as the figure caption. Please correct.
7. Figure 3, the x-axis is temperature in Celsius not kelvin. Please correct. Also, although the authors have measured the samples from 20 – 900 °C, why do you start the figure from 300 °C?
8. The conclusion section isn’t so informative; therefore, it is highly recommended to rewrite again.
9. All the references should be revised very well. The references must be uniformly formatted. Many references are written with many mistakes. See ref. 2, 3, 5 and 9 for example.
Round 2
Reviewer 4 Report
The authors have improved the manuscript, therefore, it should be accepted in the current form.